# Anti-Inflammatory, Antioxidative, and Nitric Oxide-Scavenging Activities of a Quercetin Nanosuspension with Polyethylene Glycol in LPS-Induced RAW 264.7 Macrophages

**DOI:** 10.3390/molecules27217432

**Published:** 2022-11-01

**Authors:** Sang Gu Kang, Gi Baek Lee, Ramachandran Vinayagam, Geum Sook Do, Se Yong Oh, Su Jin Yang, Jun Bum Kwon, Mahendra Singh

**Affiliations:** 1Department of Biotechnology, Institute of Biotechnology, Life and Applied Sciences, Yeungnam University, Gyeongsan 38541, Korea; 2Department of Biology, College of Natural Sciences, Kyungpook National University, Buk-gu, Daegu 41566, Korea; 3Nova M Healthcare Co., Ltd., 16-53, Jisiksaneop 4-ro, Gyeongsan 38408, Korea

**Keywords:** quercetin, nanosuspension, antioxidant properties, flavonoids, LPS-induced inflammation, NO-scavenging activity

## Abstract

Quercetin (Qu) is a dietary antioxidant and a member of flavonoids in the plant polyphenol family. Qu has a high ability to scavenge reactive oxygen species (ROS) and reactive nitrogen species (RNS) molecules; hence, exhibiting beneficial effects in preventing obesity, diabetes, cancer, cardiovascular diseases, and inflammation. However, quercetin has low bioavailability due to poor water solubility, low absorption, and rapid excretion from the body. To address these issues, the usage of Qu nanosuspensions can improve physical stability, solubility, and pharmacokinetics. Therefore, we developed a Qu and polyethylene glycol nanosuspension (Qu-PEG NS) and confirmed its interaction by Fourier transform infrared analysis. Qu-PEG NS did not show cytotoxicity to HaCaT and RAW 264.7 cells. Furthermore, Qu-PEG NS effectively reduced the nitrogen oxide (NO) production in lipopolysaccharide (LPS)-induced inflammatory RAW 264.7 cells. Additionally, Qu-PEG NS effectively lowered the levels of COX-2, NF-κB p65, and IL-1β in the LPS-induced inflammatory RAW 264.7 cells. Specifically, Qu-PEG NS exhibited anti-inflammatory properties by scavenging the ROS and RNS and mediated the inhibition of NF-κB signaling pathways. In addition, Qu-PEG NS had a high antioxidant effect and antibacterial activity against *Escherichia coli* and *Bacillus cereus*. Therefore, the developed novel nanosuspension showed comparable antioxidant, anti-inflammatory, and antibacterial functions and may also improve solubility and physical stability compared to raw quercetin.

## 1. Introduction

Inflammation is a complex biological response of body tissues against infection by pathogens, injury, or irritants, such as toxins and UV, and a repair system for damaged tissues [1]. During the inflammatory phase, macrophages secrete cytokines and chemokines, process antigens in acute immune responses and phagocytosis, and play an essential role in wound healing [2]. Macrophages in the inflammation phase regulate inflammation through the production of various inflammatory mediators, including prostaglandins, such as cyclooxygenase 2 (COX 2), nitric oxide (NO), tumor necrosis factor-α (TNF-α), interleukin-6 (IL-6), and interleukin-1β (IL-1β) [3]. Thus, COX-2, TNF-α, IL-6, and IL-1β of cytokines are biomarkers for inflammation responses [4].

Hence, inflammation is the cause of many diseases, and a variety of drugs (steroids and nonsteroidal anti-inflammatory drugs) are utilized to treat it. Hence, phytochemicals, such as polyphenols (e.g., salicylic acid and derivatives, acetylsalicylic acid (aspirin)), are commonly utilized to treat inflammation. In addition, plant polyphenols extracted from medicinal plants and their synthetic derivatives have been historically used for treating inflammatory diseases [5]. Recently, many medicinal plants enriched with polyphenols have been established for anti-inflammatory disease treatments, including *Eucalyptus globules*, *Thymus vulgaris* [6], *Mentha longifolia* [7], *Pycnocycla spinosa* [8], *Echium amoenum* (borage), and *Mitrephora sirikitiae* [9,10].

Moreover, reactive oxygen (ROS) and reactive nitrogen species (RNS) are products of metabolism that use oxygen in living cells and are highly reactive to cellular and mitochondrial membranes and lead to cell damage. ROS are the derivatives of oxygen, such as superoxide radicals (O_2_^•−^), hydroxyl radicals (OH^•^), singlet oxygen (^1^O_2_), and hydrogen peroxide (H_2_O_2_). RNS are nitric oxide (^•^NO), peroxynitrite (ONOO^−^), and other harmful chemical agents [11]. Excess ROS can oxidize biomolecules and modify proteins and genes, which can lead to the progression of inflammatory diseases [12]. Therefore, rapid clearance of ROS and RNS by antioxidants is essential to prevent the occurrence of inflammation. For rapid clearance of oxidative stresses, plant polyphenols are excellent scavengers for free radicals, such as ROS and RNS. Thus, the antioxidant-rich flavonoids of polyphenols found in vegetables and fruits lower inflammation and reduce the risk of cardiovascular and brain disease. In addition, flavonoids have the advantage of being less toxic and can be prescribed for an extended duration. Therefore, various plant-derived flavonoids use as drugs have been reported as a modulator for chronic inflammation caused by virus infection and other diseases, such as human papillomavirus [13,14], hepatitis virus [15], SARS-CoV-2 [16,17], autoimmune disease [18,19], type 2 diabetes [20,21], cardiovascular diseases [22], Alzheimer’s disease [23], Parkinson’s disease [24], and cancer [25]. The therapeutic effect of these flavonoids and polyphenols is to inhibit the function of the enzymes as ligands that attach to the specific sites of the enzymes. For example, the catechins of flavonoids have been demonstrated to inhibit tyrosinase by binding to the enzyme’s active site in molecular docking and inhibition assays [26]. Therefore, flavonoids and polyphenols are of great value in the development of disease treatments.

Quercetin (Qu) (3,3′,4′,5,7-pentahydroxy-2-phenylchromen-4-one), a plant flavonol of polyphenols, is found in grains, fruits, and vegetables and in higher levels in capers, buckwheat seeds, radish, onions, apples, red leaf lettuce, and asparagus [27,28,29]. Qu has been used as an immuno-protective and anti-inflammatory activity [30], antioxidant, antidiabetic, anticarcinogenic agent [31,32,33], antimicrobial activity [34], and ability to prevent various chronic disorders [35]. Hence, it can be taken as a supplement with the daily consumption of a nutraceutical with doses ranging from 10 to 125 mg/serving [36]. However, because quercetin possesses poor water solubility, low absorption, and fast excretion, it has shown relatively short bioavailability compared to the other nutraceutical polyphenolics [37,38]. Indeed, low bioavailability of Qu in human plasma has been observed after oral administration of Qu [38]. In addition, the low solubility and fast excretion of Qu results in minimal absorption in the gastrointestinal tract. In addition, quercetin is metabolized and lost by the gut microbiome before absorption in the gastrointestinal tract [17]. To overcome these factors, nanosuspension technology can improve the efficacy of drugs by changing the solubility, bioavailability, and pharmacokinetics [38].

Therefore, we proposed a novel Qu nanosuspension to improve solubility, physical stability, biosafety, and bioavailability. Furthermore, the Qu nanosuspensions were examined for their physicochemical characteristics, particle size, differential scanning calorimetry, cytotoxicity, anti-inflammatory, antibacterial, and antioxidant effects. The chemical structures of quercetin and polyethylene glycol are shown in the Figure 1.

## 2. Results and Discussion

### 2.1. Preparation of Nanosuspension and Physical Observation

The solubility of Qu in water is relatively low (about 60 mg/L). However, Qu is soluble in organic solvents, such as ethanol, dimethyl formamide (DMF), methanol, or acetone [39]. For the preparation of Qu nanosuspension, PEG 8000 was selected as a suitable material because of its high-water solubility, excellent stability at room temperature, and non-hygroscopic properties. In addition, Qu precipitates in aqueous acidic solutions while being soluble in aqueous alkaline solutions. By taking advantage of Qu’s poor solubility in acidic aqueous solutions, the same nanoprecipitation process was employed to produce the Qu-PEG nanosuspension. When 1 M HCl was added to the alkaline solution, Qu precipitated with the PEG 8000. The principle of nanosuspension predicted that the precipitated quercetin molecules interact with and are entrapped within the PEG 8000 molecules, resulting in the formation of quercetin nanosuspension (Qu-PEG NS). To prevent Qu from degrading, the nanoprecipitation was performed right after Qu was solubilized [40]. Hence, the final pH of nanosuspension was kept at 6–7 to prevent the Qu degradation. The final product was sonicated, lyophilized, and stored in a light-resistant air-tight container for further use.

Three different Qu-PEG NS samples were prepared (Table 1). The samples showed the different characteristics in their lyophilized form. The Qu-PEG NS1 was a completely dry and free-flowing powder. However, Qu-PEG NS2 had some lumps and was not dry. Qu-PEG NS3 had more lumps than Qu-PEG NS2 and was not completely dry under set conditions. This may be due the high quantity of PEG. In comparison to pure quercetin, Qu-PEG NS1 had better powder characteristics. Furthermore, the agglomeration was not observed in Qu-PEG NS1 as compared to Qu, as shown in Figure 2. The color of lyophilized formulations was found to be different than pure quercetin and Qu-PEG NS1 showed a red-orange color with respect to pure quercetin (yellow) (Figure 2).

Previously, several formulation technologies for nanosuspension have been developed, i.e., nanocrystals using high-pressure homogenization [41], nanosuspensions by high-pressure homogenization [42], and nanosuspension by a solvent displacement method followed by solvent evaporation [43]. In this study, quercetin nanosuspension was successfully developed to improve the solubility and stability of natural quercetin molecules using an acid-base nanoprecipitation method (Figure 2 and Table 1).

### 2.2. Determination of Content Uniformity

Approximately 94.6 ± 1.78% of the quercetin was found to be present in the freeze-dried formulation (Qu-PEG NS). The loss that may have occurred during preparation and lyophilization could be responsible for the loss of quercetin content [44].

### 2.3. ATR-FTIR Analysis

In conjunction with conventional infrared spectroscopy, attenuated total reflection (ATR) sampling allows materials to be directly viewed in solid or liquid conditions without needing further preparation. Hence, attenuated total reflectance-Fourier transform infrared (ATR-FTIR) assessment was performed to examine the interaction between the components used in the formation of nanosuspensions. The ATR-FTIR spectra of the components and their nanosuspensions (Qu-PEG NS) are shown in Figure 3.

Pure Qu has three distinct peaks at 1671, 1609, and 1511 cm^−1^, which correspond to the compound’s benzene ring and -C=O group. The FTIR spectrum of pure Qu showed evidence of stretching OH groups at 3407 and 3275 cm^−1^ and bending OH groups of the phenol at 1350 cm^−1^. At 1671 cm^−1^, the C-O aryl ketonic stretch absorption was visible. Stretch bands for the C-C aromatic ring were observed at 1609, 1558, and 1511 cm^−1^. Furthermore, peaks at 931, 816, and 600 cm^−1^ indicate the out-of-plane bending bands. Moreover, bands at 1243, 1199, and 1161 cm^−1^ reveal the C-O stretching in the aryl ether ring, the C-O stretching in phenol, the C-CO-C stretching and bending in ketone, respectively. The FTIR spectrum for pure Qu is shown in Figure 3, where its characteristic groups were detected as previously reported [45].

The spectra of PEG 8000 exhibited peaks at 1467, 1359, 1341, 1280, and 1240 cm^−1^, indicating alkyl CH deforming, as well as 1113 cm^−1^ for C-O-C stretching, 1060 cm^−1^ for C-OH stretching, and 2884 cm^−1^ for alkyl CH stretching (Figure 3), which were found similar to previously reported findings [46]. In the spectrum of PEG 8000, both trans-planar and helical structures are characteristic bands. The peaks at 1341 cm^−1^ (CH_2_ wagging), 1240 cm^−1^ (CH_2_ twisting), and 960 cm^−1^ (CH_2_ rocking) were found due to the trans-planar structure. Meanwhile, the bands of the helical structure were found at 1359 cm^−1^ (CH_2_ wagging), 1279 and 1240 cm^−1^ (CH_2_ twisting), 960 and 841 cm^−1^ (CH_2_ rocking), and 1059 cm^−1^ (coupled C-O and C-C stretching), as previously reported [47]. Other bands in PEG 8000 at 1146, 1097, and 1059 cm^−1^ can be assigned to ether groups. The physical mixture (Qu + PEG8000) showed peaks similar to Qu (Figure 3), indicating no interaction between Qu and PEG 8000. Additionally, it was observed that Qu-PEG NS indicated the broadening of peaks and showed peaks at 3310 cm^−1^ and 1635 cm^−1^; hence, this indicates that the interaction between Qu and PEG 8000 occurred, as shown in Figure 3. A previous study also demonstrated that quercetin–PEG interaction in a solid dispersion system is indicated by the broadening and shifting of the hydroxyl vibration band of PEG from 3800 to 3000 cm^−1^ [48].

### 2.4. DSC Analysis

DSC is a thermal analysis apparatus that computes the heat flow and temperature association with material conversions as a function of temperature and time. Figure 4A,B show the thermal behavior of pure Qu against the thermogram of Qu-PEG NS. It is visible that the thermogram of pure Qu revealed an endothermic peak corresponding to a swift and sharp disintegration at 324.97 °C. As seen from the DSC thermogram of the Qu-PEG NS, the peak of Qu at 324.97 °C was reduced and shifted to 211.58 °C (Figure 4A,B). An endothermic peak for Qu at 324.97 °C reveals that it exists in crystalline form. According to a previous study, quercetin’s melting point was found to be 326 °C [49] and our finding also indicates an almost similar melting point. The endothermic peak disappearance in the prepared Qu-PEG NS suggests that Qu may be molecularly diffused or be interacting with the PEG matrix. Furthermore, this endothermic peak drop suggests that Qu-PEG NS could be amorphous [50,51]. Non-crystalline substances are known as amorphous forms since they have no long-range arrangement. Furthermore, amorphous materials display an apparent second-order phase transition so-called “glass transition temperature,” or Tg, when examined using conventional thermal analytical techniques, such as differential scanning calorimetry (DSC). In this method, the temperature range is significantly below the melting point of the crystalline material. Hence, ‘Tg’ is one of a few distinguishing characteristics of an amorphous material that can be utilized to predict its suitability and stability to determine if it is appropriate for use in dosage forms [52].

### 2.5. Particle Size Measurements

The particle sizes of nanosuspensions without and with lyophilization were measured (Figure 5). All lyophilized nanosuspensions increased as compared to non-lyophilized samples. Liquid nanosuspensions showed particle sizes of 261.2 to 415.5 nm while lyophilized samples showed particle sizes in the 271.1 to 422.6 nm range. The Qu-PEG NS3 showed a smaller particle size than Qu-PEG NS1 and Qu-PEG NS2. This may be due to the increased solubility of quercetin due to the addition of higher amounts of PEG 8000. The quercetin PEG nanosuspension Sample 1 (Qu-PEG NS1) (Figure 5) was selected and named Qu-PEG NS for further study due to its free-flowing properties and particle size, which can be considerable for activity. The solid particles in nanosuspensions typically have a particle-size distribution of less than one micron, with an average particle size of 200–600 nm [53]. In this experiment, we also agreed that the nanosuspension modifies the drug’s pharmacokinetics with efficacy and safety by improving low solubility and bioavailability.

### 2.6. Antioxidant Activity

The hydrogen-donating ability of antioxidants is essential for predicting the physiological function of substances. The antioxidant function of the Qu-PEG NS was investigated to predict its physiological function (Figure 6). At a concentration of 250 µg per ml, the Qu-PEG NS displayed 74.80% of DPPH radical-scavenging activity, whereas ascorbic acid and Qu showed 82.20% and 76.40%, respectively (Figure 6A). The Qu-PEG NS showed an apparent ability to function as a free radical scavenger for DPPH inhibition, comparable to ascorbic acid, which is recognized as a free radical scavenger. The free-radical-scavenging activity of Qu-PEG NS may be attributed to the free-radical-scavenging activity of its central substance, quercetin. The antioxidant effect of quercetin is due to the phenolic hydroxyl group of quercetin. Phenolic hydroxyl groups in plant phenolic compounds can provide hydrogen to reduce free radicals and prevent the oxidation of proteins, lipids, and DNA [41]. Thus, Qu-PEG NS exhibited the same antioxidant capacity as Qu, because the amount of free hydroxyl groups contained in Qu-PEG NS is directly related to the scavenging ability of flavonoids, such as quercetin. It has been found that quercetin shows concentration-dependent antioxidant activity and, at higher concentrations (10 µg/mL), exhibits a plateau phase [54]. Our results also showed the plateau phase at a higher concentration, as shown in Figure 6A. Similarly, amylose–quercetin complexes [55] and electrospun zein nanofibrous encapsulating quercetin–cyclodextrin inclusion complex [56] showed the antioxidant activity against a DPPH radical agent.

Moreover, we also employed the ABTS assay to establish the synthetic complex’s anti-radical capability. After reacting with ABTS for 12–14 h in the dark, potassium persulphate produced the blue chromophore known as ABTS. Qu-PEG NS effectively scavenged ABTS radicals, such as a powerful antioxidant ascorbic acid (Figure 6B). Moreover, the ABTS radical-scavenging abilities of Qu-PEG NS and Qu were almost the same. Thus, although the physical composition of Qu-PEG NS differs from that of pure Qu, and Qu-PEG NS has almost the same physiological properties as Qu, that may be because it retains the powerful antioxidant function of Qu substances. Previous studies also confirmed that Qu and its formulations had favorable DPPH and ABTS antioxidant activity [57,58,59,60]. Likewise, bovine serum albumin nanoparticle promoted the stability of quercetin and decreased (*p* < 0.05) the ABTS radical-scavenging activity of Qu [61].

Additionally, Qu is widely known for its potent anti-free-radical properties and for acting as a chelating agent to inactivate the metal iron that is responsible for the production of reactive oxygen species [62]. Therefore, Qu shows antioxidant or pro-oxidant effects depending on the concentration, which is known as hormetic characteristic of Qu (biphasic dose response), where a low dose of Qu exhibits an antioxidant effect and a high dose shows a pro-oxidant effect [63,64]. Because of the hormetic properties, it is a strong candidate for investigations on cancer prevention [65]. Furthermore, it has been found that Qu is a significant and effective antioxidant and can be thought of as a powerful anticancer agent [63].

Additionally, ROS’ oxidative damage to intracellular organelles and macromolecules causes inflammation, leading to various diseases, such as heart disease, diabetes, Alzheimer’s disease, Parkinson’s disease, and cancer [12,66,67]. Therefore, we predicted that Qu-PEG NS and Qu scavenge free radicals from the cells, thus inhibiting inflammation, which helps to prevent the occurrence of chronic diseases occurrence.

### 2.7. Cell Viability of Qu and Qu-PEG NS on RAW 264.7 Cells

The commonly used MTS assay is an essential parameter to determine the degree of toxicity and to measure cell viability for new substances. HaCaT and RAW 264.7 cells were treated with Qu and prepared nanosuspension to measure cell viability by MTS assay (Figure 7A,B). The cell viability on HaCaT cells was performed with Qu and Qu-PEG NS at 5, 10, 50, and 100 μg/mL concentrations (Figure 7A). HaCaT cells treated with Qu and Qu-PEG NS at a concentration of 10 μg/mL showed cell viability of 85.2% and 89.4% compared to control cells that were not treated with anything. When the concentration of the test sample was increased by 10 folds (100 μg/mL), the survival rate was slightly lower in Qu-PEG NS (76.9%), while pure quercetin showed a significant difference in cell cytotoxicity (63.8%) at a high concentration. Therefore, Qu-PEG NS had a safer function for cells than the original native Qu molecules.

In addition, we measured the cell viability of Qu and Qu-PEG NS in RAW 264.7 cells with LPS treatment and DMSO to investigate their function on inflammation. Qu showed cell viability of 79.1% and 54.7% at concentrations of 10 and 100 μg/mL, respectively (Figure 6B), while the Qu-PEG NS showed cell viability of 80.8% and 74.8% at concentrations of 10 and 100 μg/mL, respectively. For the results of the MTS assay, the Qu and Qu-PEG NS did not show serious cytotoxicity to HaCaT and RAW 264.7 cells. Therefore, two molecules may not have caused toxicity to mouse and human normal cells. In addition, Qu-PEG NS showed less cytotoxicity than Qu at the 100 μg/mL concentration (Figure 7), suggesting that prepared nanosuspension is more biocompatible than the pure Qu molecules. When a material exhibits the anticipated favorable tissue response, clinically significant performance is called to be a biocompatible material. The other factors are cytotoxicity, mutagenicity, genotoxicity, carcinogenicity, and immunogenicity, which check the biocompatibility of the prepared material. In this study, the cytotoxicity study was performed and nanosuspension showed less cytotoxicity than the pure Qu. Hence, the prepared nanosuspension can be more biocompatible than the pure Qu.

### 2.8. The Effect of Qu and Qu-PEG NS on NO Production by RAW 264.7 Cells

Nitric oxide (NO) is a highly reactive free radical and an important secondary messenger that mediates the inflammatory response [68]. Furthermore, LPS increases the NO production in macrophages and triggers an inflammatory response [69]. Hence, we investigated the effects of Qu-PEG NS and Qu on lowering the NO production in LPS-induced macrophages. The RAW 264.5 cells were pre-treated with 10 or 100 μg/mL of Qu, Qu-PEG NS, and catechin for 48 h. Then, the RAW 264.7 cells stimulated inflammation by treatment of LPS (1 μg/mL) for 24 h and were assayed for NO levels (Figure 8). Macrophages are one of the main cell types that demonstrate the NF-κB activation in inflammatory disorders and one of the leading NO generators in vivo [70,71].

Findings were confirmed that NO synthesis in RAW 264.7 cells induced the inflammatory response by LPS treatment (Figure 8). As compared to LPS treatment alone, RAW 264.7 cells treated with Qu, Qu-PEG NS, and catechins along with LPS showed a reduction in the NO production. Therefore, Qu and Qu-PEG NS dramatically reduced the NO production of macrophage cells in inflammatory conditions (Figure 8). LPS enhances the NO synthesis during the inflammatory response by activating NF-κB and up-regulating inducible nitric oxide synthase (iNOX) [72]. Hence, we prove the anti-inflammatory properties by considering this reduction in NO generation in inflammatory macrophage cells by Qu and nanosuspension.

### 2.9. Differentiation of RAW 264.7 Cells with Qu and Qu-PEG NS in the Presence of LPS

Macrophages represent monocytes, which are highly differentiated cellular phenotypes that coordinate host inflammatory responses and wound healing. Monocyte cells exist in a gradient along this maturation pathway, depending on the external environment, from immature to mature phenotype [73]. The degree of change from the immature to mature phenotype of monocytes treated with nanosuspension was measured (Figure 9). The RAW 264.7 cells are monocytic progenitors and have a smaller, rounded phenotype with less cytoplasmic expansion before LPS treatment, as shown in Figure 9. After treatment with LPS, cells were matured to macrophages after 48 h. As a result of the morphological transformation experiment, LPS promotes adhesion, growth, and proliferation of monocytic cells to maturate and spread the culture surface. This indicates that LPS was modified to induce an inflammatory response in the immature RAW 264.7 cells. The RAW 264.7 cells were incubated with relative amounts of Qu or Qu-PEG NS with 1 µg/mL of LPS and recorded by phase contrast microscopy (Figure 8). After 24 h of Qu and Qu-PEG NS treatment, RAW 264.7 cells had many untransformed monocytes. On the other hand, 24 h after treatment with Qu and Qu-PEG NS with LPS, RAW 264.7 cells became mature macrophages. This indicates that LPS matures macrophages due to an inflammatory response. Therefore, macrophages treated with Qu or Qu-PEG NS remained immature monocytes until LPS treatment (24 h) because the quercetin and its nanosuspension did not induce an inflammatory response. LPS is a gram-negative bacterial cell wall component and causes inflammatory bone loss by converting macrophages into osteoclasts [74]. Furthermore, LPS induces the production of various cytokines and mediators, such as tumor necrosis factor (TNF)-α, interleukin (IL)-1, and prostaglandin E2 (PGE2), in macrophages and plays a vital role in the maturation of macrophages [36]. Here, we confirmed that LPS-induced RAW 264.7 cells were transformed into macrophages which involves in the inflammatory response. Since Qu and Qu-PEG NS were decreased the NO production (Figure 8). Hence, they lowered the inflammatory response in LPS-induced inflammatory RAW 264.7 cells.

### 2.10. Effect of Qu and Qu-PEG NS on Inflammatory Proteins in LPS-Induced RAW 264.7 Cells

Cytokines, including TNF-α, IL-6, and IL-1β, are pro-inflammatory [75]. Furthermore, IFN-γ and/or LPS-stimulated macrophages that have produced TNF-α are synergistically induced to produce NO [76]. Numerous physiological reactions are induced by TNF-α, including septic shock, cachexia, inflammation, and cytotoxicity [77]. In addition to controlling the expression of pro-inflammatory cytokines and enzymes, including iNOS, COX-2, TNF-α, and IL-6, NF-kB is a recognized biomarker of cellular inflammation responses [78].

Therefore, we analyzed the inflammation-related effects of Qu and Qu-PEG NS by increasing or decreasing inflammatory biomarkers, such as IL-1β, COX-2, and TNF-α-activated NF-κB (p65), by Western blot analysis (Figure 10).

In the absence of LPS, the COX-2 protein was not expressed, while IL-β1 and NF-κB were constitutively expressed (Figure 10A). However, when Qu and Qu-PEG NS were used, the IL-β1 and NF-κB were significantly down-regulated compared to β-actin (Figure 10A). While in the presence of LPS, the RAW 264.7 cells induced the COX-2 protein, indicating inflammation (Figure 10B). Furthermore, expression of COX-2, IL-β1, and NF-κB were markedly up-regulated by LPS. However, pretreatment with Qu and Qu-PEG NS of 100 µg/mL significantly reduced the levels of COX-2, IL-β1, and NF-κB as compared to β-actin level (Figure 10B).

When LPS stimulates the cells, NF-κB is transferred to the nucleus to regulate gene expression [79]. Here, the reduction in NF-κB may result in down-regulation of COX-2, IL-1β, and NO production levels, indicating an anti-inflammatory effect (Figure 8 and Figure 10). Additionally, it has been previously observed that quercetin successfully inhibited Nod-like receptor family pyrin domain-containing 3 (NLRP3) of the inflammasome family [29]. Inflammasomes are cytoplasmic multiprotein oligomers that generate active forms of cytokines IL-1β and IL-18 to activate the inflammatory response [80]. Therefore, by weakening the inflammasome function, quercetin is a potential treatment for severe inflammation, such as SARS-CoV-2-induced cytokine storm and Alzheimer’s disease, and life-threatening diseases, such as atherosclerosis and cardiac arrhythmias. Generally, ROS-induced transcription factors, including NFκB, and pro-inflammatory factors, such as COX-2, TNF-α, IL-6, and IL-1β, lead to the onset of inflammation [8]. As with our experimental results, a high antioxidant effect of quercetin and epigallocatechin gallate (EGCG) reduces ROS production to inhibit thioredoxin-interacting protein (TXNIP)-NLRP3 inflammasome and decrease IL-1β production [81]. The possible mechanisms of inflammation by LPS and anti-inflammatory action by Qu and Qu-PEG NS are summarized in Figure 11. A possible mechanism of anti-inflammation of Qu and Qu-PEG NS may be due to high antioxidant ability, which could be responsible for reducing the activation of NLRP3 inflammasome, thereby reducing cytokines. In addition, Qu and Qu-PEG NS have a high ROS-/RNS-scavenging ability that inhibits the NF-κB pathway to reduce iNOS, COX-2, and lipoxygenase expression levels, leading to anti-inflammatory action.

### 2.11. Antibacterial Activity

Pathogenic bacteria can infect humans and cause fatal diseases. The current significant issue of modern medicine is the development of drugs for drug-resistant bacteria. Though antibiotics are the first line of protection against bacterial illness, the evolution of pathogenic drug resistance has encouraged the design of highly efficient and new antimicrobial medications [82]. Antimicrobial resistance is a global problem because it induces antibiotic resistance and a rise in healthcare expenses [83]. Hence, effective antibacterial materials are considerably required. Even though many species from the *Bacillus* and *Escherichia* genera are beneficial for food fermentations, probiotics, and industrial purposes [84,85], many of these species are also referred to as pathogens. *B. cereus* is a facultative, anaerobic, gram-positive, and spore-forming soil, plant, and food bacterium, which has been associated with the development of foodborne diseases, including severe nausea, vomiting, and diarrhea [86]. Additionally, *E. coli* is a facultative, anaerobic, gram-negative, rod-shaped, coliform bacteria typically detected in mammals’ gastrointestinal tract [87]. While most *E. coli* strains are not harmful to humans, others, such as *E. coli* O157:H7 has been linked to severe food poisoning [88]. Therefore, the anti-microbial activity of Qu-PEG NS against these bacteria has emphasized the potential non-antibiotic treatments. Hence, in this investigation, we used colony-forming unit (CFU) methods to assess the antibacterial activity of and Qu-PEG NS at different concentrations against *B. cereus* and *E. coli*. As shown in Figure 12, the CFU of *B. cereus* was reduced to 171 × 10^6^ when treated with Qu-PEG NS (0.1 mg/mL) while 0.5 mg/mL and 1 mg/mL did not show any growth in the number of bacteria as compared to 342 × 10^6^ CFU/mL for the untreated control. However, Qu showed 139 × 10^6^ CFU/mL, 109 × 10^6^ CFU/mL, and 42 × 10^6^ CFU/mL at concentrations of 0.1, 0.5, and 1.0 mg/mL, respectively. The Qu-PEG NS treatments (0.5, 1.0 mg/mL) also significantly inhibited *B. cereus* growth compared to Qu and control (Figure 12).

*E. coli* treated with 0.1 and 0.5 mg/mL of Qu-PEG NS showed a CFU of 126 × 10^7^ CFU/mL and 40 × 10^7^ CFU/mL, respectively (Figure 13). The 1.0 mg/mL treatment of Qu and Qu-PEG NS did not produce any colonies compared to 129 × 10^7^ CFU/mL for the untreated control. The antimicrobial activity of Qu and Qu-PEG NS may be due to microbial cell lysis by precipitation of membrane proteins of bacteria [89]. At a dosage of 0.5 mg/mL, the Qu-PEG NS was found to be more efficient against gram-positive bacteria, such as *B. cereus,* compared to gram-negative bacteria. This result may be related to the possibility that the nanosuspension can easily adhere to the bacterial membrane and induce cell lysis through protein precipitation. Thus, Qu-PEG NS can be used as a non-antibiotic antimicrobial agent against gram-positive bacterial infections.

## 3. Materials and Methods

### 3.1. Chemicals and Cell Culture

HaCaT and RAW 264.7 (KTCC No. 40071) cells were taken from the Korea Cellular Bank (KTCC, Seoul, Korea). Penicillin (100 units/mL)/streptomycin (100 g/mL) and fetal bovine serum (FBS), 0.25% trypsin-EDTA, Cell Titer 96 Aqueous one solution cell proliferation assay kit (Prommega, Madison, WI, USA), lipopolysaccharide (Sigma Aldrich, St. Louis, MO, USA), Halt Protease Inhibitor Cocktail, EDTA-Free (Thermo Scientific, Waltham, MA, USA), Goat Anti-Rabbit IgG antibody (HRP) (GeneTex, Irvine, CA, USA), m-IgGκ BP-HRP (Santa Cruz Biotechnology, Dallas, TX, USA), M-PER^®^ Mammalian Protein Extraction Reagent (Thermo Scientific Waltham, MA, USA), Western Enhanced Buffer (NEOSCIENCE, Seoul, Korea), Super Signal^®^ West Pico Chemical Substrate (Thermo Scientific Waltham, MA, USA), PVDF membrane (Sigma Aldrich, St. Louis, MO, USA), phosphate buffer saline (PBS) (Welgene, Gyeongsan, Republic of Korea; Sigma, St. Louis, MO, USA), sodium hydroxide, hydrochloric acid, polyethylene glycol 8000, quercetin, and other reagents (Sigma Aldrich) were used.

### 3.2. Preparation of Nanosuspension

Nanosuspension was prepared by the acid-base nanoprecipitation method. First, a mixture of quercetin and polyethylene glycol 8000 (0.6 g) was weighed in different ratios such as 2:1, 1:1, and 1:2. The weighed quantity of combinations was dissolved separately in 20 mL of 0.2 N NaOH solution at room temperature using a magnetic stirrer (Dathan scientific, MSH-20D) at 500 rpm to form a clear solution. Then, 3.5 mL of 1M HCl was added under continuous magnetic stirring at 1500 rpm for 15 min to obtain the precipitated quercetin suspension. Furthermore, the prepared suspension pH was adjusted to 6–7 with 0.1 N NaOH or 0.1M HCl. Then, the formulation was sonicated with a probe sonicator (Vibra cell sonicator, 750-watt model) for 20 min at 35% amplitude at room temperature to formulate nanosuspensions. Then, the prepared nanosuspensions were frozen at −70 °C for 24 h and lyophilized at −60 °C for 48 h. The lyophilized samples were stored in an air-tight container for further evaluation.

#### 3.2.1. Physical Observation and Determination of Content Uniformity

The Qu and lyophilized nanosuspensions (Qu-PEG NS1, Qu-PEG NS2, and Qu-PEG NS3) were examined for agglomeration and color using stereomicroscopy. The samples were scattered on a glass slide and focused and the photomicrographs were taken.

The amount of quercetin in the freeze-dried product was determined by dissolving 10 mg of lyophilized nanosuspension in 10 mL of methanol. After 15 min of probe sonication and filtering with a 0.22 µm membrane filter, the absorbance of quercetin was measured spectrophotometrically at 370 nm. The concentration of quercetin was calculated using the regression equation (y = 0.0673x + 0.0375, r^2^ = 1), which was obtained by preparing the standard curve of quercetin in methanol (where ‘y’ is the absorbance of the test sample and ‘x’ is the concentration to be determined). Among the three types of nanosuspension in Table 1, Qu-PEG NS1 was selected and named Qu-PEG NS to investigate the physiological activity properties.

#### 3.2.2. Analytical Characterization

Attenuated total reflectance-Fourier transform infrared (ATR-FTIR) spectroscopy was recorded on a Perkin Elmer (Waltham, MA, USA) in the infrared region (4000 and 600 cm^−1^) and analyzed by transmittance technique at a spectral resolution of 1 cm^−1^.

The DSC curves of the different samples were recorded on a differential scanning calorimeter (Perkin Elmer Inc., Waltham, MA, USA) calibrated with indium. The thermal behavior was studied by heating samples in aluminum crimped pans under nitrogen gas flow. The samples were heated from −10 °C to 340 °C at 10 °C/min to obtain the thermogram.

#### 3.2.3. Particle Size Measurements

Quercetin nanosuspensions were analyzed for particle size by photon correlation spectroscopy (PCS) using a Zetasizer Nano ZS (Zen 3600, Malvern Instruments, Worcestershire, United Kingdom). The instrument scientifically and instinctively adjusts to the test sample by altering the intensity of the laser and the attenuator of the photomultiplier, thus confirming the reproducibility of the experimental measurement conditions.

### 3.3. Determination of Antioxidant Capacity

The DPPH˙ free-radical-scavenging activity with Qu and Qu-PEG NS was evaluated in contrast to ascorbic acid [87]. In brief, 200 µL of the 10, 50, 100, 250, 500, and 1000 µg/mL NQS1 solution and 100 µL of the 0.2 mM DPPH solution in ethanol were combined. The mixture was then left to react for 30 min at room temperature before the absorbance at 517 nm was recorded using an ELISA reader. Equations (1) and (2) were used to determine the DPPH˙ free-radical-scavenging activity.
DPPH scavenging activity % = [1 − Absorbance sample/Absorbance control] × 100(1)

As previously reported, the ABTS^+^ radical-cation-scavenging method was performed [40]. To generate ABTS^+^ radicals, 7 mM ABTS and 2.5 mM potassium persulfate were combined in an equal volume and left at room temperature in the dark for 24 h. Then, using an ELISA reader (Infinite^TM^ F200, Männedorf, Switzerland), the absorbance of this solution was measured at 734 nm after being diluted twice with ethanol. An ELISA reader was used to measure the absorbance at 734 nm to check the scavenging activity of Qu and Qu-PEG NS in an equal volume (150 µL) of ABTS solution and with different concentrations of test samples, i.e., 10, 50, 100, 250, 500, and 1000 µg/mL of Qu and Qu-PEG NS. The reaction was allowed to proceed for 6 min at room temperature. ABTS scavenging activity was calculated by the following equation.
ABTS scavenging activity % = [1 − Absorbance sample/Absorbance control] × 100(2)

### 3.4. Nitric Oxide (NO) Production Measurement

After treatment with LPS, NO production assay was performed to measure the effect of Qu and Qu-PEG NS on NO production from RAW 264.7 cells. The RAW 264.7 cells were uniformly dispensed in 24 wells at a density of 2 × 10^5^ cells/well and then each test solution was applied at a concentration of 10 or 100 μg/mL and incubated for 48 h. Furthermore, cells were treated with LPS (1 μg/mL) and incubated for 96 h (LPS treatment for 48 h). After that, 50 μL of the cell supernatant was collected and reacted with 50 μL of the sulfanilamide solution (1% sulfanilamide in 5% phosphoric acid) for 10 min. Then, 50 μL of the NED solution (0.1% N-1-naphthylenediamine) was used to react for 10 min, and the absorbance was measured at 548 nm by utilizing an ELISA reader. A standard curve of sodium nitrite was utilized to calculate the NO concentration.

### 3.5. Measurement of Cell Viability of Qu and Qu-PEG NS

Cytotoxicity and proliferation rates of Qu and Qu-PEG NS on HaCaT and RAW 264.7 cells were measured after the treatment with lipopolysaccharide (LPS, 1μg/mL). Testing procedures were performed as previously reported, with some modifications to detect cell cytotoxicity [90]. The HaCaT and RAW 264. 7 cells were evenly distributed in a 96-well plate at a density of 1 × 10^4^ cells/well to measure cytotoxicity and proliferation. After 24 h, each sample was treated with a solution containing 10 or 100 µg/mL and incubated for 96 h. The impact of concentration-specific therapy on cell proliferation was examined using the Cell Titer 96 Aqueous one solution cell proliferation assay kit. All the media were taken out after incubation and then 20 µL of DMEM medium and 100 µL of 3-(4,5-dimethylthiazol-2-yl)-5-(3-carboxymethoxyphenyl)-2-(4-sulfophenyl)-2H-tetrazol inner salt (MTS) solution were added. The sample optical density was measured with an ELISA reader using a 96-well plate (InfiniteTM F200, Switzerland). The absorbance at 390 nm, 37 °C, and 3 °C was measured. As a control, a culture solution without any treatment was employed.

### 3.6. Anti-Inflammatory Activity of LPS-Induced RAW 264.7 Cells with Qu and Qu-PEG NS

In a 100 mm petri plate, macrophages (RAW 264.7 cells) were cultured at a density of 4.5 × 10^7^ cells/plate for 24 h. Afterward, cells were rinsed with PBS and treated with various concentrations (10 and 100 µg/mL) of quercetin and nano-quercetin and incubated for 48 h, followed by cells being exposed to LPS (1 μg/mL) and further incubated for 96 h to induce the inflammation. Then, the protein was extracted from the sample-treated RAW 264.7 cells using M-PER^®^ Mammalian Protein Extraction Reagent and Halt Protease Inhibitor Cocktail, EDTA-Free, and the protein content was assessed.

Proteins extracted from treated RAW 264.7 cells were used for Western blot analysis. Briefly, 30 μg of protein was separated by electrophoresis using 7.5% sodium dodecyl sulfate-polyacrylamide gel (SDS-PAGE) and afterward transferred to PVDF membrane (polyvinylidene fluoride) and blocked with Western Enhanced Buffer (NEOSCIENCE, Seoul, Korea). The primary antibodies were employed in a ratio of 1:2000 and 1:1000. The secondary antibodies were Goat Anti-Rabbit IgG antibody (HRP) (GeneTex, Alton Pkwy Irvine, CA, USA) and m-IgGκ BP-HRP (Santa Cruz Biotechnology, Dallas, TX, USA). The primary antibodies COX-2, IL-1β NF_k_B p65, and β-actin were purchased from GeneTex or Santa Cruz Biotechnology. The secondary antibodies were labelled in the ratio of 1:10000. After completing the second antibody reaction, a Super Signal^®^ West Pico Chemical Substrate (Thermo Scientific, Waltham, MA, USA) solution was applied to a PVDF membrane (Sigma Aldrich, St. Louis, MI, USA), made photosensitive to film in a darkroom, and developed. The relative strength of a specific protein band was determined utilizing an Image J^®^ analyzer (National Institutes of Health, Bethesda, MD, USA).

### 3.7. Antimicrobial Activity

*Bacillus cereus* (ATCC 14579) and *Escherichia coli* (ATCC 15597) were purchased from the ATCC (American Type Culture Collection, Manassas, VI, USA) and assessed for the antibacterial activity with Qu and Qu-PEG NS (0.1 and 1.0 mg/mL). Bacterial culture was performed in LB (Luria–Bertani) media for 12 h at 37 °C with shaking at 120 rpm. Tetracycline was used as a positive control. Bacteria (10^4^ cells/mL) were then collected and washed with PBS (pH 6.8). These cells were then added to fresh LB media with varied sample concentrations and cultured at 37 °C for another 12 h with shaking. The sample-treated bacterial cultures were serially diluted from 10^−5^ to 10^−7^ and then plated on LB agar plates and incubated at 32 °C for 12 h. The antibacterial activity of the Qu and Qu-PEG NS was determined by colony counting on the culture plates.

### 3.8. Statistical Analysis

All experiments were repeated three times and an average ± standard deviation represented statistical analysis. The student’s t-test method was used (GraphPad 8 trial version) and *p* < 0.05 indicated a statistically significant difference.

## 4. Conclusions

According to physicochemical characterization and the acid-base condition of the synthesis, we addressed the development of nanosuspensions. In addition, this study observed that prepared nanosuspension showed favorable biosafety against cell line studies. Furthermore, the nanosuspension significantly protected RAW 264.7 cells by regulating NO overproduction and ROS levels. Moreover, nanosuspension showed suitable anti-inflammatory activity by reducing the COX-2, IL-1β and NF-κB levels along with anti-bacterial activity. Hence, it can be concluded that the prepared nanosuspension possesses considerable antioxidant, anti-inflammatory, and antibacterial activity, which may help to prevent the occurrence of chronic diseases.

## Figures and Tables

**Figure 1 molecules-27-07432-f001:**
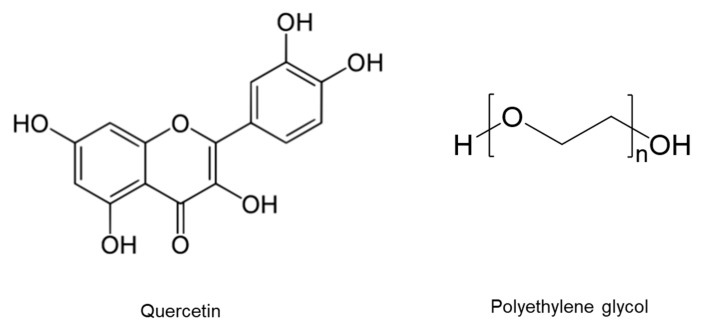
Chemical structures of quercetin and polyethylene glycol.

**Figure 2 molecules-27-07432-f002:**
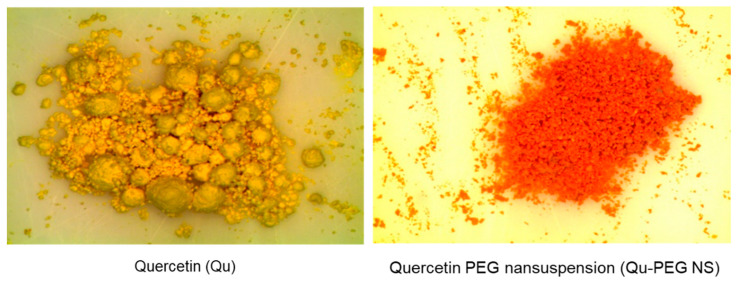
Stereomicroscopy photographs of quercetin (Qu) and quercetin with PEG nanosuspension 1 (Qu-PEG NS).

**Figure 3 molecules-27-07432-f003:**
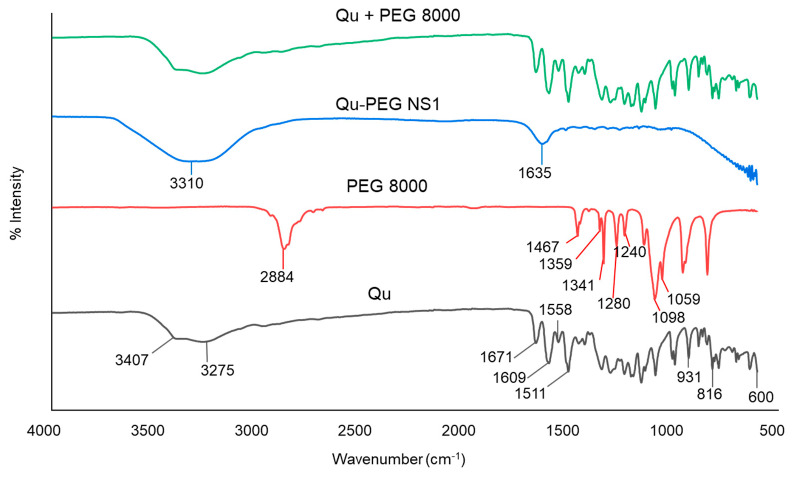
Attenuated total reflectance-Fourier transform infrared (ATR-FTIR) calculation of Qu, PEG 8000, Qu-PEG NS, and Qu + PEG 8000.

**Figure 4 molecules-27-07432-f004:**
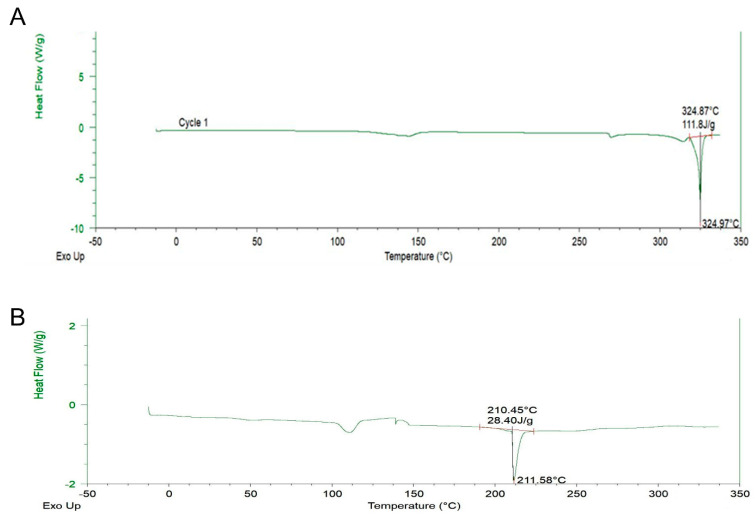
Differential scanning calorimeter (DSC) thermograms of (**A**) Qu and (**B**) Qu-PEG NS.

**Figure 5 molecules-27-07432-f005:**
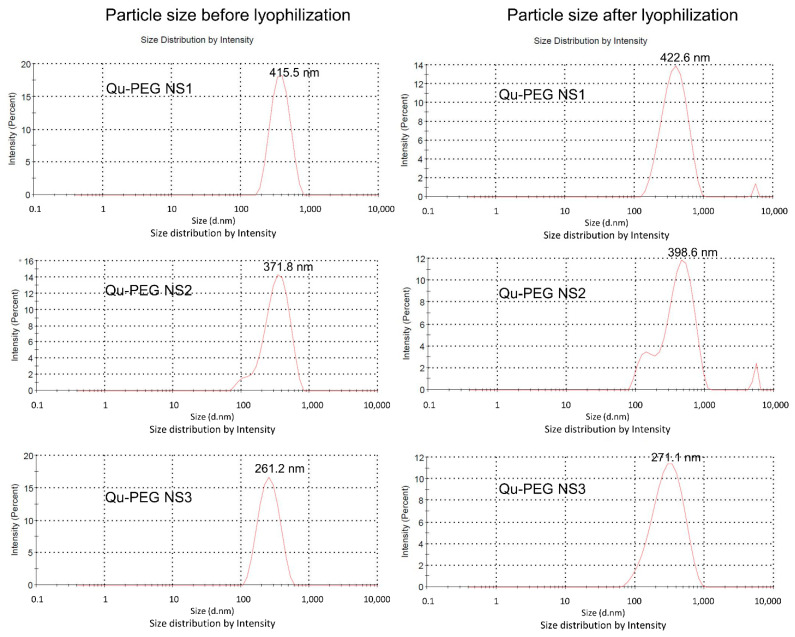
Particle-size distribution of nanosuspensions before and after lyophilization.

**Figure 6 molecules-27-07432-f006:**
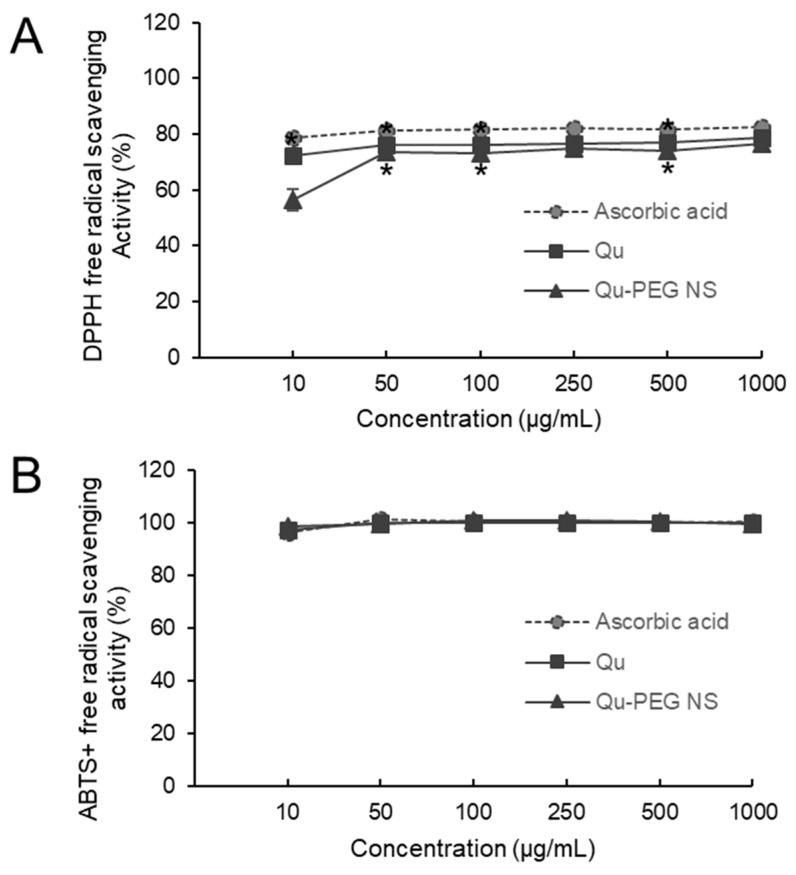
Effect of Qu and Qu-PEG NS on (**A**) DPPH and (**B**) ABTS free radical scavenging, * *p* < 0.05 indicates significant difference as compared to control.

**Figure 7 molecules-27-07432-f007:**
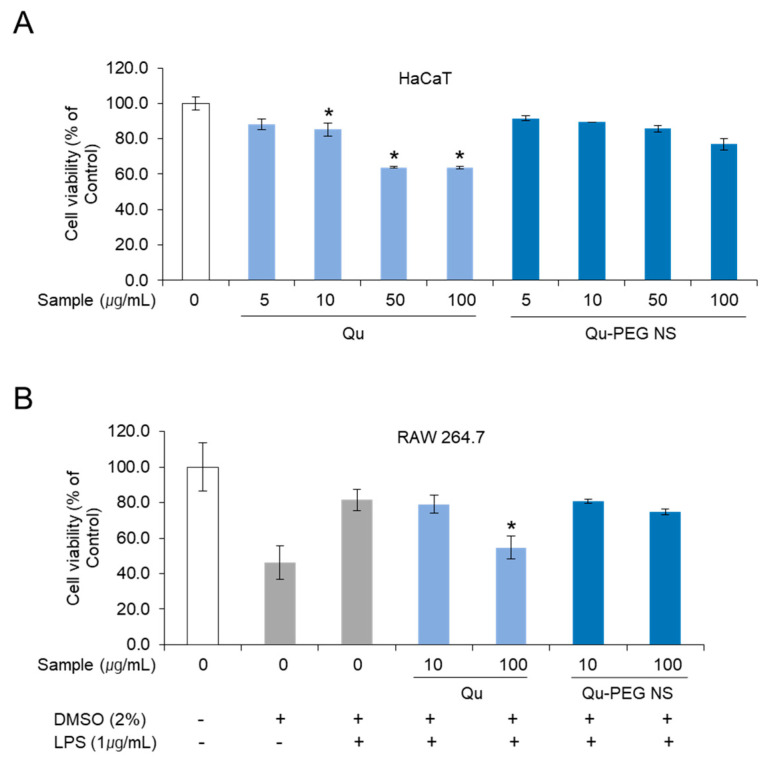
MTS analysis of (**A**) HaCaT and (**B**) RAW 264.7 cell viability treated with Quercetin and Qu-PEG NS. The cells were pre-treated with different concentrations of quercetin. Qu-PEG NS was then treated with LPS (1μg/mL) for 48 h. Results are the mean ± S.D. of samples. * *p* < 0.05 indicates a significant difference from the Qu-PEG NS.

**Figure 8 molecules-27-07432-f008:**
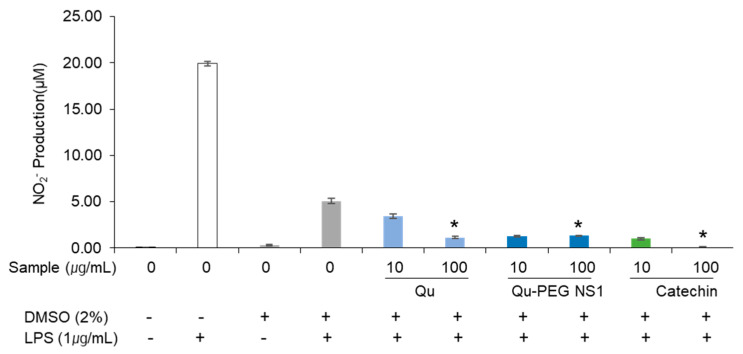
Effects of Qu and Qu-PEG NS on NO production by RAW 264.7 cells in the presence of LPS. Results are the mean ± SD of test samples, * *p* < 0.05 indicates a significant difference from the control at a concentration of 100 μg/mL.

**Figure 9 molecules-27-07432-f009:**
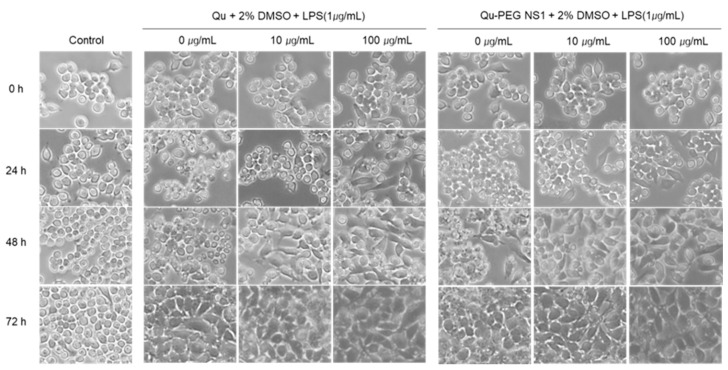
The morphological changes in LPS-induced RAW 264.7 cells treated with Qu and Qu-PEG NS. The RAW 264.7 cells were incubated with Qu or Qu-PEG NS for 24 h (24 h) and then LPS was added and grown (48 h and 72 h). Phase contrast microscopy was used to capture the images of RAW 264.7 (150× magnification).

**Figure 10 molecules-27-07432-f010:**
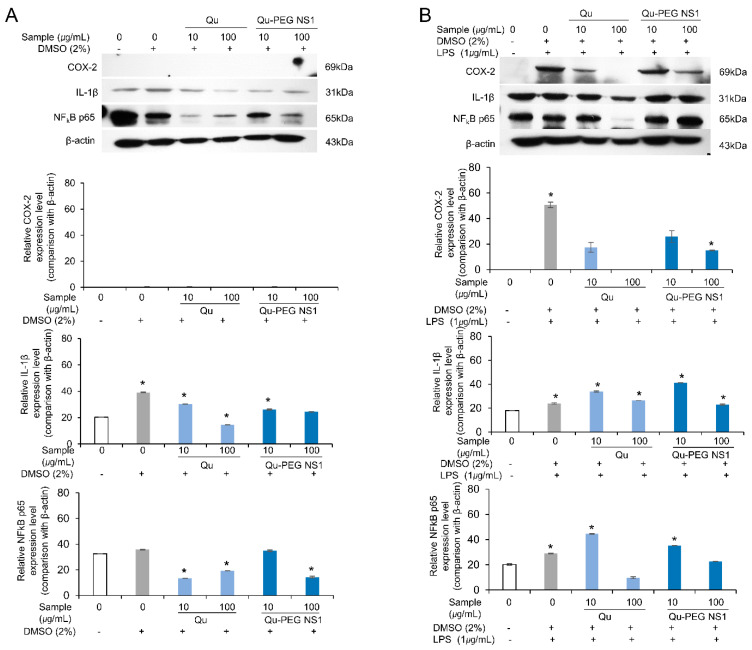
Effects of Qu and Qu-PEG NS on RAW 264.7 cells (**A**) in the absence of LPS and (**B**) in the presence of LPS on the expression of COX-2, IL-1β, and TNF-α-activated NF-κB (p65) as biomarkers of inflammation. The β-actin was employed as base level of protein expression. * *p* < 0.05 indicates a significant difference.

**Figure 11 molecules-27-07432-f011:**
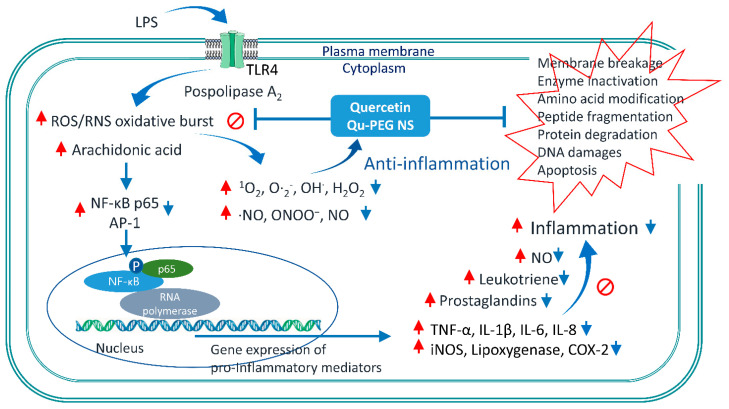
Schematic representation of the mechanism of Qu and Qu-PEG NS anti-inflammation effects in LPS-induced inflammatory RAW 264.7 cells. In this study, the Qu and Qu-PEG NS quench ROS and RNS, which may be responsible for inhibiting the active NF-κB transcription factor and suppression iNOS, Cox-2, and pro-inflammatory mediator protein levels, including TNFα, IL-1β, IL-6, and IL-8. Furthermore, the attenuation of NF-κB activity accompanies the inhibition effects of Qu and Qu-PEG NS by preventing NF-κB translocation from the cytoplasm to the nucleus. Red arrows indicate for inflammation pathway by LPS. Blue arrows indicate the anti-inflammation pathway by Qu or Qu-PEG NS molecules. Abbreviations: Activator protein 1 (AP-1), Cytochrome c oxidase subunit 2 (COX-2), inducible nitric oxide synthases (iNOS), Interleukin 1 beta (IL-1β), Interleukin 6 (IL-6), Interleukin 8 (IL-8) Nuclear factor kappa-light-chain-enhancer of activated B cells (NF-κB), Nuclear factor NF-kappa-B p65 subunit (NF-κB p65), reactive nitrogen species (RNS), reactive oxygen species (ROS), toll-like receptor 4 (TLR4), lipopolysaccharide (LPS), tumor necrosis factor alpha (TNF-α).

**Figure 12 molecules-27-07432-f012:**
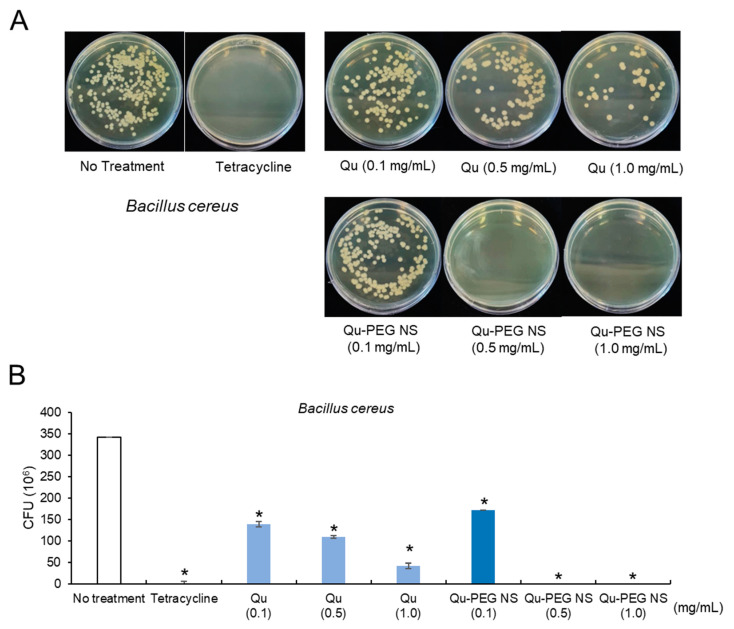
Antibacterial activity of Qu and Qu-PEG NS against *B. cereus* (**A**) Colony forming on growth medium and (**B**) Number of colonies forming units, * *p* < 0.05 significant difference compared to control.

**Figure 13 molecules-27-07432-f013:**
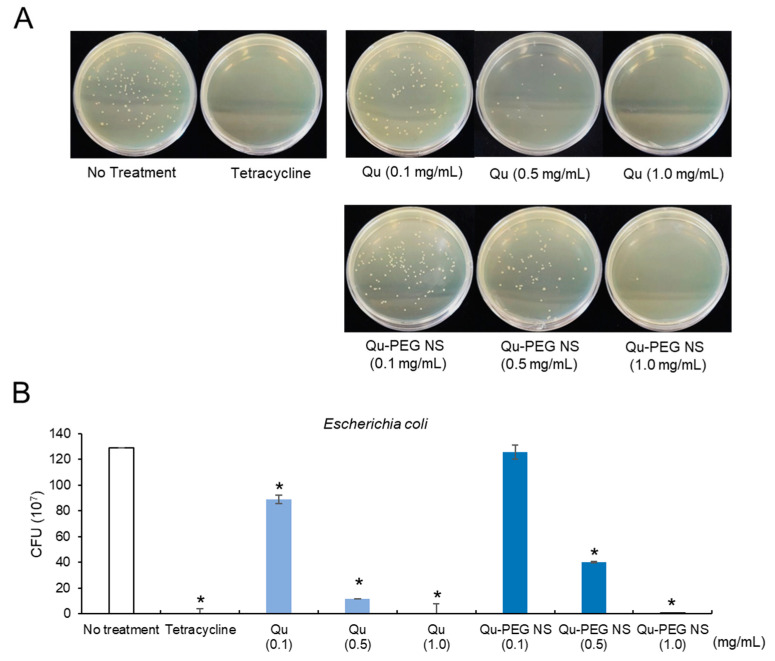
Antibacterial activity of Qu and Qu-PEG NS against *E. coli* (**A**) Colony forming on growth medium and (**B**) Number of colonies forming units, * *p* < 0.05 significant difference compared to control.

**Table 1 molecules-27-07432-t001:** The composition ratio of nanosuspension formulation.

Formulation	Ratio (Quercetin: PEG8000)
Qu-PEG NS1	2:1
Qu-PEG NS2	1:1
Qu-PEG NS3	1:2

## Data Availability

Not applicable.

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
