# Peer review of "Anti-Inflammatory, Antioxidative, and Nitric Oxide-Scavenging Activities of a Quercetin Nanosuspension with Polyethylene Glycol in LPS-Induced RAW 264.7 Macrophages"

_molecules, 2022, doi:10.3390/molecules27217432_

Round 1

Author Response

Response to Reviewer’s Comments

Manuscript title:  Anti-inflammatory, anti-oxidative and nitric oxide-scavenging activities of a quercetin nanosuspension with polyethylene glycol in LPS-induced RAW264.7 macrophages

Journal Name: Molecules, MDPI

Manuscript No: Molecules-1985328

Author(s): Sang Gu Kang, Gi Baek Lee, Ramachandran Vinayagam, Geum Sook Do, Se Yong Oh, Su Jin Yang, Jun Bum Kwon, and Mahendra Singh 

Dear Editor

We would like to thank you for your time and effort in reviewing our manuscript and suggesting corrections to improve the quality of our manuscript. We have incorporated the reviewer`s suggestions.

The following is a point-by-point response to the reviewer's comments. We mark all revised items in red in the recent version according to the reviewer's suggestions. In addition, the authors revised the manuscript, and the revisions are marked in red. English editing is completed.

Reviewer 1

Review of manuscript number Molecules- 1985328

Title: Anti-inflammatory, anti-oxidative, and nitric oxide-scavenging activities of a quercetin nanosuspension with polyethylene glycol in LPS-induced RAW264.7 macrophages

The article describes the results of the investigation on the preparation of nanosuspension to improve the stability of quercetin. The authors performed three parts of the initial experiment by preparation of formulations called Qu-PEG NS1, Qu-PEG NS2, and QuPEG NS3. The analysis carried out in this study were physicochemical characteristics (particle size measurements, DSC), antioxidant, anti-inflammatory, and antibacterial properties. Some parts of the paper should have been changed. Moreover, the English language needs improvement.

Query 1: Keywords: please specify other words than antioxidant, antimicrobial, anti-inflammatory Do you mean antioxidant properties?

RESPONSE: Thank you for your pertinent suggestion. Authors have added some new keywords as per suggestions and highlighted in the revised manuscript.

Query 2: Introduction

This paragraph is too long, could the authors make it shorter?

RESPONSE: Thank for valuable suggestion. The authors have been made the corrections in the revised manuscript.

Query 3: Material and methods

2.2.1. Can the authors explain how many samples were analyzed? And how the analysis was performed?

RESPONSE: Thank you for your valuable query. Authors have done the corrections in the revised manuscript.

Query 4: 2.3 Why authors chose ascorbic acid to perform such an analysis? The standard substance used in this analysis is Trolox. Then, it is easier to compare your results with other scientists results.

RESPONSE:  Thank you for your pertinent comment and suggestion. Ascorbic acid was available with us, so used as a standard antioxidant. In future authors will keep in mind to perform the antioxidant activities of test sample against Trolox.

Query 5: 2.8. Did the authors use only one statistical test?

RESPONSE: Authors were used only one statistical test analysis to compare the results.

Query 6: Results and discussion, 3.2. How authors calculated the content of quercetin in the freeze-dried formulations? How the quercetin content was established? In the materials and methods paragraph, there was any method of such determination presented.

RESPONSE: Thank you for your pertinent query. Authors were drawn the standard curve of quercetin and calculated the regression equation for calculation of the concentration of quercetin in the freeze-dried sample. Authors have been made the corrections and highlighted in the revised manuscript.

The concentration of quercetin was calculated using the regression equation (y= 0.0673x+0.0375, r2 =1) which was obtained by preparing the standard curve of quercetin in methanol. Where ‘y’ is the absorbance of the test sample and ‘x’ is the concentration to be determined.  

Query 7: 3.4 Can authors explain what kind of amorphous of quercetin can be found in nature?

RESPONSE: Non-crystalline substances are known as amorphous forms since they have no long-range arrangement. Furthermore, amorphous materials display an apparent second-order phase transition so-called "glass transition temperature," or Tg, when examined using conventional thermal analytical techniques such as differential scanning calorimetry-DSC. In this, the temperature range is significantly below the melting point of the crystalline material. Hence, ‘Tg’ is the one of a distinguishing characteristics of an amorphous material that can be utilized to predict its suitability and stability if it is appropriate for use in dosage forms [1].

Query 8: 3.6 Please add some other data to discuss your result with other scientists results.

RESPONSE: It has been found that quercetin shows concentration dependent antioxidant activity and at higher concentration (10 µg/mL) exhibits plateau phase [2] and our results were also showed the plateau phase at higher concentration as shown in Figure 6A. Furthermore, amylose/quercetin complexes [3], and electrospun zein nanofibrous encapsulating quercetin/cyclodextrin inclusion complex [4] were showed the antioxidant activity against DPPH radical agent.

Previous studies also confirmed that Qu and their formulations had good DPPH and ABTS antioxidant activity [5-8]. Likewise, bovine serum albumin nanoparticle promotes the stability of quercetin and decreased (p < 0.05) the ABTS radical scavenging activity of Qu [9].

Query 9: 3.7 How did you manage to suggest that your nanomaterial was more biocompatible than pure quercetin? please explain it in this part of the article.

RESPONSE: Thank you for the valuable query. Authors have been done the corrections in the revised manuscript.

When a material exhibits the anticipated favorable tissue response and clinically significant performance is called to be a biocompatible material. The other factors are cytotoxicity, mutagenicity, genotoxicity, carcinogenicity, and immunogenicity to check the biocompatibility of the prepared material. In this study, cytotoxicity study was performed and nanosuspension showed the less cytotoxicity than the pure Qu. Hence, prepared nanosuspension can be more biocompatible than the pure Qu.

Query 10: Figure 11. Did the authors prepare this figure all by themselves?

RESPONSE: Yes, authors were prepared the Figure 11 by themselves.

Query 11: Conclusion: This section is unclear, please rewrite it once again, because the sentences in this paragraph should be more like statements and show the positive side of your results experiments. The last sentence of this section is not true because the paper does not present experiments made on humans.

RESPONSE:

According to physico-chemical characterization and the acid-base condition of the synthesis addressed the development of nanosuspensions. Also, this study observed that prepared nanosuspension was showed good biosafety against cell line studies. Further-more, the nanosuspension was significantly protected RAW 264.7 cells by regulating NO overproduction and ROS levels. Moreover, nanosuspension was showed good an-ti-inflammatory activity by reducing the COX-2, IL-1β and NF-κB levels along with an-ti-bacterial activity. Hence, it can be concluded that the prepared nanosuspension pos-sesses considerable antioxidant, anti-inflammatory, and antibacterial activity which may help to prevent the occurrence of chronic diseases.

Reviewer 2 Report

This is interesting and relevant work.

1) 2.2.1. physical observation. This item should be described in more detail, or not stand out as a separate item.

2) The paper should discuss the possible pro-oxidant effects of high doses of quercetin. As well as possible pro-oxidant effects of quercetin when added during oxidative stress.

3) It would be nice to arrange all the figures in color

Author Response

Reviewer 2

This is interesting and relevant work.

Query 1) 2.2.1. physical observation. This item should be described in more detail, or not stand out as a separate item.

RESPONSE: Thank for the valuable suggestion. Author have combined the section 2.2.1 and 2.2.2 into one section 2.2.1. Physical observation and determination of content uniformity in the revised manuscript.

Query 2) The paper should discuss the possible pro-oxidant effects of high doses of quercetin. As well as possible pro-oxidant effects of quercetin when added during oxidative stress.

RESPONSE: Qu is widely known for its potent anti-free radical properties and for acting as a chelating agent to inactivate the metal iron that is responsible for the production of reactive oxygen species [10]. Since, Qu shows antioxidant or pro-oxidant effect depends on the concentration, which is known as hormetic character of Qu (biphasic dose response), where low dose of Qu exhibits antioxidant effect and high dose shows the pro-oxidant effect [11, 12]. Because of the hermetic properties it is a strong candidate for investigations on cancer prevention [13]. Furthermore, it has been found that Qu is a significant and effective antioxidant and can be thought of as a powerful anticancer agent [11].

Query 3) It would be nice to arrange all the figures in color

RESPONSE: Thank you for your valuable suggestion. Authors have been made the corrections in the revised manuscript.
